# Antimicrobial Susceptibility and Genetic Prevalence of Extended-Spectrum β-Lactamases in Gram-Negative Rods Isolated from Clinical Specimens in Pakistan

**DOI:** 10.3390/antibiotics12010029

**Published:** 2022-12-24

**Authors:** Muhammad Mubashar Idrees, Rimsha Rimsha, Muhammad Daoud Idrees, Ali Saeed

**Affiliations:** 1Institute of Molecular Biology and Biotechnology, Bahauddin Zakariya University, Multan 60800, Pakistan; 2Multan Institute of Kidney Diseases (MIKD), Multan 60800, Pakistan; 3Department of Medical Laboratory Technology, Faculty of Medicine and Allied Health Sciences, The Islamia University of Bahawalpur, Bahawalpur 63100, Pakistan; 4National Institute of Food Science and Technology, University of Agriculture, Faisalabad 38000, Pakistan; 5Department of Biochemistry, Bahauddin Zakariya University, Multan 60800, Pakistan; 6Department of Pediatric Oncology & Medical Microbiology, University Medical Center Groningen, University of Groningen, P.O. Box 72, 9700 AB Groningen, The Netherlands

**Keywords:** extended-spectrum β-lactamases (ESBLs), gram-negative bacteria, CTX-M, TEM, SHV, OXA, disk diffusion method, polymerase chain reaction (PCR)

## Abstract

The prevalence of extended-spectrum β-lactamase (ESBL) genes has increased remarkably, resulting in multidrug-resistant gram-negative rods (GNRs) in clinical specimens. This cross-sectional study aimed to determine the antimicrobial susceptibility of ESBL-producing GNRs and its correlation with corresponding genes. Two hundred and seventy-two (*n* = 272) samples were evaluated for the molecular identification of ESBL genes by polymerase chain reaction after confirmation with the modified double-disc synergy test. *E. coli* 64.0% (*n* = 174) was the most prevalent ESBL producer, followed by *Klebsiella* species 27.2% (*n* = seventy-four), *Acinetobacter* species 6.6% (*n* = eighteen) and others 2.2% (*n* = six). These ESBL-producing isolates showed resistance to β-lactam antibiotics, i.e., sulbactam/cefoperazone (41.5%), piperacillin/tazobactam (39.3%), meropenem (36.0%), imipenem (34.2%) and non- β-lactam antibiotics, i.e., nalidixic acid (89.0%), co-trimoxazole (84.9%), ciprofloxacin (82.4%), gentamicin (46.3%), nitrofurantoin (24.6%), amikacin (19.9%) and fosfomycin (19.9%). The incidences of the ESBLs-producing genes *bla_CTX-M,_ bla_TEM_*, *bla_OXA_* and *bla_SHV_* were 91.2%, 61.8%, 39.3% and 17.6%, respectively. Among nine multiple-gene combinations, *bla_CTX-M_ + bla_TEM_* (30.5%) was the most prevalent combination, followed by *bla_CTX-M_ + bla_OXA_ + bla_TEM_* (14.0%), *bla_CTX-M_ + bla_OXA_* (13.6%), *bla_CTX-M_ + bla_TEM_ + bla_SHV_* (7.0%), *bla_CTX-M_ + bla_SHV_* (2.2%), *bla_CTX-M_ + bla_OXA_ + bla_SHV_* (2.2%) and *bla_OXA_ + bla_TEM_* (1.8%). ESBLs producing GNRs carrying *bla_CTX-M,_ bla_TEM_*, *bla_OXA_* and *bla_SHV_* showed resistances to β-lactam antibiotics, i.e., ampicillin, amoxillin-clavulanic acid, cefotaxime and ceftazidime but were susceptible to carbapenems (meropenem and imipenem), β-lactam-β-lactamase inhibitor combination (piperacillin/tazobactam) and non-β-lactam antibiotics i.e., aminoglycoside (amikacin and gentamicin), nitrofurantoin and fosfomycin. These antibiotics that demonstrated activity may be used to treat infections in clinical settings.

## 1. Introduction

β-lactamases are mainly plasmid-mediated enzymes produced by some gram-negative rod-shaped bacteria that cause antimicrobial resistance against different antimicrobial groups, i.e., penicillin (ampicillin), expanded-spectrum cephalosporins (cefotaxime and ceftazidime) and monobactam (aztreonam) [1,2]. β-lactamases play a vital role in resisting or protecting gram-negative bacteria (GNB) against β-lactam antibiotics by breaking their β-lactam ring, leading to the ineffectiveness of β-lactam antibiotics [3,4]. The worldwide incidence of GNB-producing extended-spectrum β-lactamases (ESBLs) has remarkably increased in urinary tract infections (UTI) both in community and clinical settings and is certainly a big therapeutic challenge [5,6,7]. Carbapenemases, AmpCs and ESBLs are more common β-lactamases produced by GNB. Originally, *bla_TEM-1_* was a plasmid-based β-Lactamase (*bla*) gene detected in the 1960s [8]. After this, a group of ESBL-coding genes were identified in *Klebsiella pneumoniae* and *Serratia marcescens* in 1983, due to a point mutation in the *bla_TEM_* and *bla_SHV_* genes [9]. Previously, about 600 types of β-Lactamases existed that were resistant to β-Lactamase inhibitors, i.e., sulbactam, clavulanic acid and tazobactam; these included *bla_SHV_*, *bla_TEM_*, *bla_OXA_* and *bla_CTX_*_-M_ [10,11]. β-Lactamases are divided into four classes based on their protein homology classes. A (TEM, SHV and CTX-M), C (AmpC) and D (OXA) are serine-based β-Lactamase and group B are metallo-β-lactamases [12]. *bla_TEM_* and *bla_SHV_* were found to be common genotypes in GNB-producing ESBLs [13]. Nevertheless, the prevalence of the *bla_CTX-M_* gene (class A ESBLs) has increased, and it is the predominant genotype found worldwide [14]. The CTX-M-based ESBLs have higher catalytic activities against cefotaxime than ceftazidime [2,15]. Strains with CTX-M-type ESBLs are commonly associated with a resistance to aminoglycosides and fluoroquinolones, but in addition, trimethoprim/sulfamethoxazole and tetracycline are resistant in strains with the SHV- and TEM-type ESBLs [16,17].

The overuse of antibiotics, inappropriate hygienic practices in clinical settings and a lack of continuous monitoring of drug-resistance patterns have led to an environment for the emergence and uncontrolled dissemination of genes encoding AmpC and ESBL enzymes [18].

It is an alarming situation in clinical settings to treat and overcome infections caused by ESBL-producing pathogens that possess resistances to commonly used antibiotics. There is a lack of awareness in both the population and health care workers about the prevalence of these pathogens and the potential ramifications to controlling these infections [19]. The incidence of ESBL-producing pathogens is tremendously increasing due to lack of proper laboratory diagnosis, reporting and preventive measures in our clinical settings [20,21]. The World Health Organization (WHO) has shown a serious concern in the increase of antibiotic resistance in developing countries, especially Pakistan, and has asked for them to overcome this issue [2]. Different research groups have conducted their studies in different regions of Pakistan [22] such as Lahore (2022) [3], Timurgara [4] and Islamabad [5], but no studies were conducted in Multan, located in South Punjab, Pakistan. Thus, the importance of this study is due to lack of epidemiological data and the prevalence of ESBL producers and their antimicrobial-susceptibility pattens in our population. In view of this, the study was conducted to find out the susceptibility profiles of ESBL-producing isolates to various antibiotics to treat infections caused by the ESBL-producing isolates. Moreover, the prevalences of the corresponding ESBL genes (*bla_CTX-M,_ bla_OXA,_ bla_TEM_* and *bla_SHV_)* among clinical isolates in the Multan Institute of Kidney Disease Hospital, Multan, Pakistan was evaluated.

## 2. Results

### 2.1. Socio-Demographic Data and Distribution of ESBL-Producing Bacterial Isolates

A total of two hundred and seventy-two (*n* = 272) ESBL-producing clinical isolates were isolated. By sample type, 72.8% (*n* = 198) were collected from urine, 6.3% (*n* = 17) from blood, 5.9% (*n* = 16) from pus, 4.8% (*n* = 13) from throat aspiration, 2.9% (*n* = 8) from CVC Tip, 1.8% (*n* = 5) from sputum, 1.5% (*n* = 4) from PCN, 1.5% (*n* = 4) from bronchial aspiration and 2.6% (*n* = 7) from others (ascitic fluid and cerebrospinal fluids). In total, 50.7% (*n* = 138) isolates were identified in males, followed by females 49.3% (*n* = 134). The prevalence of ESBLs was more abundant in out-patients (OPD) at 45.2% (*n* = 123), followed by 21.3% (*n* = 58) from nephrology, 19.1% (*n* = 52) from urology, 9.6% (*n* = 26) from emergency and 4.8% (*n* = 13) from the intensive-care unit (ICU). *Escherichia coli* at 64.0% (*n* = 174) were the most prevalent ESBL-producing bacterial isolates identified in this study, followed by *Klebsiella* species 27.2% (*n* = 74), *Acinetobacter* species 6.6% (*n* = 18) and others (*Salmonella*, *Stenotrophomonas* species, *Serratia*) 2.2% (*n* = 6) (Table 1) and detailed information are available in the Appendix A.

### 2.2. Susceptibility of ESBL-Producing Gram-Negative Rods (GNRs) to Antibiotics 

The activities of different antibiotics against GNB such as *Escherichia coli, Klebsiella* species, *Acinetobacter* species and others (*Salmonella*, *Stenotrophomonas* species, *Serratia*) were tested in this study. Resistance was the highest for nalidixic acid, followed by NOR > SXT > CIP > G > SCF > TZP > IPM > MEM > F > AK > FOS against GNRs (Table 2). The GNRs were most susceptible to the following agents from highest to lowest: FOS > AK > F > MEM > IPM > TZP > SCF > G > CIP > SXT > NOR and NA. In conclusion, ESBL producers possessed the lowest MICs for FOS, AK, F, MEM, IPM, TZP, SCF and G. Thus, these agents may potentially be treatment options. Conversely, NA, NOR, SXT and CIP showed higher rates of resistance against GNRs, so they should not be used for treatment (Table 2).

### 2.3. Susceptibility of Escherichia coli to Various Antibiotics

*E. coli* (64.0%, *n* = 174/272) was the most prevalent bacterial species found among the tested ESBL-producing clinical isolates, and its susceptibility to antibiotics was as follows: FOS > AK > F > MEM > IPM > TZP > SCF > G > CIP > SXT > NOR and NA. The resistance rates of *E. coli* against antibiotics were NA > NOR > SXT > CIP > G > SCF > TZP > IPM > MEM > F > AK > FOS. The most effective drugs against ESBLs producing *E. coli* were FOS, AK, F, MEM, IPM, TZP and SCF and G. NA, NOR, SXT and CIP were not effective (Table 2).

### 2.4. Susceptibility of Klebsiella Species to Various Antibiotics

*Klebsiella* species (27.0%, *n* = 74/272) were the second-most-prevalent organism identified in the tested ESBL-producing bacteria after *E. coli*. The effectiveness rates of antibiotics against *Klebsiella* species were as follows: FOS > AK > IPM, TZP > G > MEM, SCF > F > NOR > CIP > SXT, NA. FOS and AK were the most effective drugs against these ESBL-producing *Klebsiella* species (Table 2).

### 2.5. Susceptibility of Acinetobacter Species and Others to Various Antibiotics

*Acinetobacter* species (6.6%. *n* = 18) were the least-prevalent organisms as compared with *E. coli* and *Klebsiella* species. *Acinetobacter* species showed high rates of resistance against all antibiotics used in this study.

A few strains of *Salmonella*, *Stenotrophomonas* species and *Serratia* (2.2%, *n* = 6) were identified among the ESBLs producing bacteria, and they were more susceptible to AK, IPM, TZP, MEM, SCF, F, FOS and G. These isolates were more resistant to SXT, CIP, NOR and NA in this study (Table 2).

### 2.6. Prevalence of bla_CTX-M,_ bla_OXA,_ bla_TEM_ and bla_SHV_ among GNRs

*bla_CTX-M_* (91.2%, *n* = 248/272) was the most prevalent gene coding for ESBL production among clinical isolates, followed by *bla_TEM_* (61.8%, *n* = 168/272), *bla_OXA_* (39.3%, *n* = 107/272) and *bla_SHV_* (17.6%, *n* = 48/272). This study revealed that *bla_CTX-M_* was the most significant gene among *E. coli, Klebsiella* species, *Acinetobacter* species and others, followed by *bla_TEM_*, *bla_OXA_* and *bla_SHV_* (Table 3).

### 2.7. Correlation between Antibiotic Resistance and ESBL Genes 

Two hundred and seventy-two isolates were phenotypically positive for ESBLs; 270 were confirmed by PCR to have ESBL-coding genes alone or in combination and two were negative. The most common gene combinations among clinical isolates were *bla_CTX-M_* and *bla_TEM_* (*n* = 83/270), followed by various combinations of *bla_CTX-M_*, *bla_TEM_*, *bla_SHV_* and *bla_OXA_*. All ESBLs-positive isolates showed resistance to β-lactam drugs (AMP, AMC, CTX and CAZ). In addition to β-lactam antibiotics, fluoroquinolones antibiotics (NA, NOR and CIP) were not effective against pathogens that possessed either *bla_TEM_*, *bla_SHV_*, *bla_CTX-M_* & *bla_SHV_*, *bla_OXA_* & *bla_TEM_*, *bla_TEM_* & *bla_SHV_*, *bla_CTX-M_* & *bla_OXA_* & *bla_TEM_*, *bla_CTX-M_* & *bla_OXA_* & *bla_SHV_*, *bla_CTX-M_* & *bla_TEM_* & *bla_SHV_*, *bla_OXA_* & *bla_TEM_* & *bla_SHV_* or *bla_CTX-M_* & *bla_TEM_* & *bla_OXA_* & *bla_SHV_* (Table 4). There was not a single clinical isolate that possessed both the *bla_OXA_* and *bla_SHV_* genes.

## 3. Discussion

The presence of ESBL-producing *Enterobacteriaceae* has emerged as a critical problem on a global scale. Infections caused by these multidrug-resistant organisms are associated with significant fatality rates and a limited number of treatment choices [23,24]. Many GNB produce ESBL enzymes that can hydrolyse cephalosporins and penicillins; clavulanic acid can inhibit them [25]. Several ESBL-producing GNRs are also multi-drug resistant to non-β-lactam antibiotics, including co-trimoxazole, nalidixic acid, norfloxacin, ciprofloxacin and amikacin. Resistance genes are often encoded by the same plasmids necessary to produce ESBL [26]. Our study revealed that *E. coli* (*n* = 174/272, 64%) was the most prevalent ESBLs-producing organism, followed by *Klebsiella* species (*n* = 74/272, 27.2%), *Acinetobacter* species (*n* = 18/272, 2.2%) and others, including *Salmonella*, *Stenotrophomonas* species and *Serratia* (*n* = 6/272, 2.2%). Our finding was confirmed by different authors who reported that *E. coli* and *Klebsiella* species were the most prevalent organism over others in Pakistan [27,28].

Our study found a high level of antibiotic resistance towards β-lactam antibiotics, including penicillin (resistance varied from 97% to 100% against amoxicillin/clavulanate and 100% against ampicillin) and cephalosporins (resistance varied from 91% to 100% against both cefotaxime and ceftazidime). Our findings were supported by different studies conducted in Argentina [29], Turkey [30], Algeria [28], Pakistan [31,32,33] and th United States [34]. The widespread and careless use of newer antibiotics, especially in the treatment of post-operative patients and in intensive care units, contributes to such high incidence of antibiotic resistance. 

Our study reported less resistance to aminoglycosides: antibiotic resistance varied from 24% to 46% for amikacin and gentamicin, while only 24% of isolates were resistant against carbapenems including meropenem and imipenem in this study. Sid Ahmed et al., 2016, reported less resistance to aminoglycoside and carbapenem antibiotics against the pathogens in their study which is in contrast with our findings, possibly due to geological and periodic variations [31]. Another study conducted by Ejaz et al., 2021, reported more resistance to carbapenems as compared with aminoglycosides [32]. Haider et al., 2022, reported higher resistance rates for both aminoglycosides and carbapenems [33]. Similarly, Merah-Fergani et al., reported of both carbapenem and aminoglycoside resistance in clinical isolates [28]. However, there is significant data inconsistency due to variances in the species investigated, antibiotic abuse, self-medication, sample methodologies and geographic locations.

Our investigation found that piperacillin/tazobactam, meropenem, imipenem, gentamicin, amikacin, nitrofurantoin and fosfomycin were the most effective antibiotics against ESBL-producing isolates. Therefore, these antibiotics likely can be used to treat infections caused by these bacteria, whereas the remaining antibiotics lacked activity and so cannot be used as treatment options. Previous research conducted by different authors found that piperacillin/tazobactam, meropenem, imipenem, gentamicin, amikacin, nitrofurantoin and fosfomycin were the most effective drugs for treating infections caused by ESBL producers [28,30,31,32,35,36,37]. Therefore, similar studies should continuously be conducted in developing countries to evaluate the performances of highly effective drugs which are commonly used to combat infections.

In this study, the molecular prevalence of ESBLs-coding genes including *bla_CTX-M,_ bla_TEM_*, *bla_OXA_* and *bla_SHV_* (99.26%), *bla_CTX-M_* (91.2%), *bla_TEM_* (61.8%), *bla_OXA_* (39.3%) and *bla_SHV_* (17.6%) among GNRs was remarkably similar to the different studies conducted in neighboring countries [38,39,40,41] and across the world [42]. The prevalences of *bla_CTX-M,_ bla_TEM_*, *bla_OXA_* and *bla_SHV_* among *Escherichia coli, Klebsiella* species, *Acinetobacter* species and others varied from 66.7% to 92.0%, 60.3% to 72.2%, 16.7% to 40.8% and 14.4% to 22.2%, respectively. According to studies conducted in Iraq and neighboring countries, the *bla_CTX-M_* gene was the dominant gene type in both *E. coli* and *Klebsiella pneumoniae* [43]. Currently, CTX-M enzymes have supplanted SHV and TEM enzymes as the predominant ESBL type in *E. coli* [44]. However, research conducted in Turkey and India revealed that TEM was the most prevalent type [45].

In this study, *bla_CTX-M_* + *bla_TEM_* (30.5%) was the most prevalent combination, followed by *bla_CTX-M_* + *bla_OXA_* + *bla_TEM_* (14.0%), *bla_CTX-M_* + *bla_OXA_* (13.6%), *bla_CTX-M_* + *bla_TEM_* + *bla_SHV_* (7.0%), *bla_CTX-M_* + *bla_SHV_* (2.2%), *bla_CTX-M_* + *bla_OXA_* + *bla_SHV_* (2.2%) and *bla_OXA_* + *bla_TEM_* (1.8%). According to the previous study conducted in Nigeria, *bla_SHV_* + *bla_TEM_* + *bla_CTX_*_-M_ (70%) was the most prevalent combination, followed by *bla_TEM_* + *bla_CTX_*_-M_ (15%), which contradicts our findings [43]. This study includes additional illustrations. Multiple genes (*bla_SHV_*, *bla_TEM_*, *bla_CTX_*_-M_) found in the genomes of some of these isolates suggested the presence of resistance plasmids [40]. Polse et al., (2016) in Iraq found six genotype patterns in a small number of samples (*n* = 50), which was exceptionally lower than our study’s (*n* = 272) 10 genotype patterns due to the small sample size [38]. Multiple ESBL genes in many ESBL-producing strains may have complex antimicrobial resistance, resulting in co-resistance to other antibiotic groups besides β-lactam antibiotics. In our study, *bla_TEM_*, a broad-spectrum ESBL, was always combined with *bla_CTX_*_-M_. According to this study, the occurrence of *bla_TEM_* + *bla_CTX-M_* + *bla_OXA_* can result in resistance to penicillins, cephalosporins and fluoroquinolones. Due to the gradual increase of resistant strains to penicillin, cephalosporins and fluoroquinolones, which were recommended and commonly used to treat infections caused by ESBL-producing bacteria, are gradually experiencing setbacks. This is both alarming and grave for community and clinical settings. This emphasizes the significance of rational antibiotic therapy, reducing the dissemination of such strains in healthcare setting and awareness of the clinical manifestation of ESBL types. Gene sequencing may also provide the phylogenetic history that will play a vital role in overcoming the further dissemination of ESBL-corresponding genes [22,46].

In conclusion, *bla_CTX-M_* was the most prevalent gene coding to ESBLs production among *E. coli, Klebsiella* species, *Acinetobacter* species and other clinical isolates, followed by *bla_TEM_*, *bla_OXA_* and *bla_SHV_*. All possible genetic combinations of ESBLs-coding genes (*bla_CTX-M,_ bla_TEM_*, *bla_OXA_* and *bla_SHV_*), except for *bla_OXA_* + *bla_SHV,_* were identified in 270 isolates of GNRs. The high rates of resistance among bacteria were due to the production of ESBLs, resulting β-lactam antibiotics (penicillins and cephalosporins) being ineffective, along with non-β-lactam antibiotics (fluoroquinolones and co-trimoxazole). Our study showed that piperacillin/tazobactam, carbapenems, aminoglycosides, nitrofurantoin and fosfomycin may be used to treat infections caused by ESBL-producing bacteria. Clinical microbiology labs should consistently use ESBL-identification tools to observe multidrug-resistant isolates and antibiograms to guide physicians and clinical staff for empirical therapy against infections. Infection prevention and control and antibiotic stewardship programs should be implemented in hospitals to limit the spread of resistant isolates.

## 4. Materials and Methods

### 4.1. Sample Collection, Transportation and Preservation

A total of two hundred and seventy-two (*n* = 272) ESBLs producing clinical isolates were isolated from urine, blood, throat aspiration, central venous catheter (CVC) tip, sputum, Percutaneous nephrostomy (PCN) urine, bronchial aspiration, ascitic fluid and cerebrospinal fluids. Samples were collected from patients admitted in different wards, i.e., out-patients door (OPD), emergency, nephrology, urology and the intensive care unit (ICU) of Multan Institute of Kidney Diseases Hospital, Multan, Pakistan (150-bed and single-specialty hospital) without age or gender discrimination in this study. Urine, sputum, fluids and PCN samples were collected in sterile containers and blood samples in culture vials (BD BACTEC™ Plus Aerobic/F, Franklin Lakes, NJ, USA) [47,48]. The samples were transported to the laboratory after collection by following different instructions, as urine samples within 2 to 4 h were placed on ice packs and blood culture vials (BD BACTEC™ Plus Aerobic/F, Franklin Lakes, NJ, USA) were held for 4 to 8 h at room temperature. After processing, urine samples were preserved at 2 °C to 8 °C in a refrigerator and blood culture vials at room temperature for one week [48].

This cross-sectional study was designed and conducted at Bahauddin Zakariya University, Multan from September 2020 to March 2021, and ethical approval was granted by the Institutional Review Board (IRB) of IMBB (Reference No. 334/A).

### 4.2. Sample Inoculation and Bacterial Identification

The urine samples were inoculated onto cysteine-lactose-electrolyte-deficient media (CLED, Oxoid, Basingstoke Hampshire, United Kingdom), and the remaining samples (throat aspiration, central venous catheter (CVC) tip, sputum, bronchial aspiration, ascitic fluid and cerebrospinal fluids) onto blood, chocolate and MacConkey agar (Oxoid, Basingstoke Hampshire, United Kingdom) plates [49]. The blood culture bottles were incubated at room temperature for 24 h followed by their subculture onto blood, chocolate and MacConkey agar. After inoculation, plates were placed at 37 °C in an incubator for overnight incubation. On the next day (after 24 h), plates were checked for bacterial growth and sub-cultured pure growth for making bacterial glycerol stock, gram staining, biochemical analysis for bacterial identification, antimicrobial susceptibility testing, and DNA extraction.

### 4.3. Bacterial Identification by Gram Staining and Biochemical Test

Pure bacterial growths were used for gram staining that was used to differentiate into gram-positive and gram-negative bacteria, either cocci or rods [50]. After confirmation of GNRs, analytical profile index 20 *Enterobacteriaceae* (API 20E, bioMérieux, Durham, NC, USA) was used to characterize the bacterial strain as *E. coli*, *Klebsiella* species, *Acinetobacter* species, *Proteus* species, *Serratia* species or *Salmonella* species [51] followed by antimicrobial susceptibility testing that was used to find which strains were resistant and susceptible to the antibiotics. Gram-positive bacteria were excluded in this study.

### 4.4. Antimicrobial Susceptibility Testing

The Kirby–Bauer disk diffusion method was used to determine the antimicrobial activity by measuring the zone of inhibition. In this study, 0.5 McFarland was used to make the inoculum by mixing bacterial colonies and normal saline in a test tube. This suspension was spread on Mueller–Hinton agar (MHA, Oxoid, Basingstoke Hampshire, United Kingdom) plates by cotton swab, following which antibiotic disks (Oxoid, Basingstoke Hampshire, United Kingdom) were dispensed on it. The zone of inhibition was measured and interpreted as resistant (R), intermediate (I) and susceptible (S) according to the guidelines given by Clinical and Laboratory Standards Institute (CLSI) [52]. The different antibiotics and their abbreviations and amounts used were as follows: ampicillin (AMP, 10 μg), amoxicillin/clavulanate (AMC, 10/20 μg), piperacillin/tazobactam (TZP, 100/10 μg), cefotaxime (CTX, 30 μg), sulbactam/cefoperazone (SCF), ceftazidime (CAZ, 30 μg), meropenem (MEM, 10 μg), imipenem (IPM, 10 μg), gentamicin (CN, 10 μg), amikacin (AK, 30 μg), nalidixic acid (NA, 30 μg), norfloxacin (NOR, 10 μg), ciprofloxacin (CIP, 5 μg), trimethoprim/sulfamethoxazole (SXT, 1.2/23.5 μg), fosfomycin (FOS, 200 μg) and nitrofurantoin (F, 300 μg) used against *Enterobacteriaceae*.

### 4.5. Phenotypic Identification of ESBLs

The Phenotypic identification of the ESBLs was carried out by the double-disk synergy test (DDST) [53]. In this study, inoculum (0.5 McFarland) was spread over a MHA agar plate by cotton swab and then amoxicillin–clavulanate was dispensed (10 μg) in-between ceftazidime (30 μg) and cefotaxime (30 μg) at a distance of 30 mm away from each other. After overnight incubation at 37 °C, a clear extension of the edge of the inhibition zone was observed in the decreased inhibition of cefotaxime or ceftazidime with a clear-cut enhanced zone of inhibition in front of the amoxicillin–clavulanate disk, referred to as ESBLs positive test [54]. 

### 4.6. Molecular Detection of ESBLs

DNA extraction was carried out by the boiling method, in which one to two bacterial colonies were emulsified into distilled water in microcentrifuge tubes and then placed in water bath at 100 °C for 20 min. Microcentrifuge tubes were centrifuged for 5 min at 3000 rpm and the supernatant that can be used as DNA was removed and the precipitate was discarded [55]. DNA was preserved at −20 °C in a deep freezer. A total of 1% agarose gel containing ethidium bromide was used to confirm the DNA extraction under UV light.

*bla_CTX-M,_ bla_OXA,_ bla_TEM_* and *bla_SHV_* were identified by PCR, in which a specific primer sequence was used, as mentioned in Table 5. The PCR was carried out using 15 μL of reaction mixture containing 1 μL (5 μM) of each forward and reverse primer, 7.5 μL of master mix (Vazyme Biotech Co., Nanjing, China), 2.5 μL of DNA (125 ng) and 3 μL of distilled water. Bio-Rad T100 Thermal Cycler (Hercules, CA, USA) conditions were adjusted as: first cycle at 94 °C for 5 min followed by 35 cycles at 94 °C for 30 s, 50 °C (*bla_CTX-M_*)/56 °C (*bla_OXA,,_ bla_TEM,_* and *bla_SHV_*) for 40 s, 72 °C for 40 s and a final extension at 72 °C for 5 min. The PCR product was analysed by using 1.2% agarose gel with staining dye (ethidium bromide) and 220 volts were applied for 40 min before visualization under UV light (Figure 1).

### 4.7. Quality Control

Different American type culture collection (ATCC) strains (Manassas, Virginia, near Washington, DC, USA) were used to check the accuracies, precisions and reproducibilities of the growth-supporting characteristics of the prepared media (CLED agar, Blood agar, Chocolate agar, MacConkey agar and MHA agar), gram staining, biochemical tests (oxidase test, catalase test, indole test, urease test, citrate test and TSI test) [49]. The accuracy of the susceptibility testing was confirmed by dispensing antibiotics against ATCC strains, and their results were interpreted according to CLSI guidelines [59]. The different strains used in this study were *Escherichia coli* (ATCC 25922), *Proteus mirabilis* (ATCC 35659), *Pseudomonas aeruginosa* (ATCC 8427) and *Staphylococcus aureus* (ATCC 25923). 

### 4.8. Statistical Analysis

Statistical analysis was carried out using the chi-square test on the data represented in this study. A tool used for statistical analysis was GraphPad Prism 9 (GraphPad Software, San Diego, CA, USA). A *p*-value ≤ 0.05 was representative of the significance of data statistically.

## Figures and Tables

**Figure 1 antibiotics-12-00029-f001:**
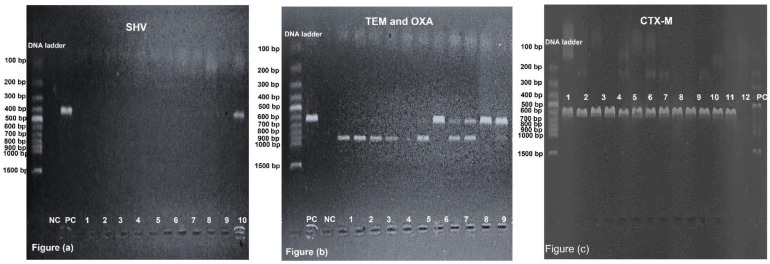
Visualization of DNA bands by gel electrophoresis. (**a**) Lane-NC and PC stand for negative and positive control, respectively. Lane-12 is showing band of 392 bp, which is related to *bla_SHV_* gene, and lane-1, -2, -3, -4, -5, -6, -7, -8, -9, -10 and -11 are negative for *bla_SHV_* gene. (**b**) Lane-NC and PC stand for negative and positive control, respectively. Lane-1, -2, -3, -4 and -6 are showing bands of 860 bp, which is related to *bla_TEM_* gene. Lane-7 and -10 are showing bands of 619 bp, which is related to *bla_OXA_* gene. Lane-8, -9 and -11 are showing both 860 bp and 619 bp bands, which are related to *bla_TEM_* and *bla_OXA_*, respectively. Lane-5 has no band. (**c**) Lane-NC and PC stand for negative and positive control, respectively. Lane-1 to -12 are showing bands of 593 bp, which is related to *bla_CTX_*_-M_ gene.

**Table 1 antibiotics-12-00029-t001:** Socio-demographic data of clinically isolated bacteria.

Age		0–9	10–19	20–29	30–39	40–49	50–59	60–69	>70
Sample Size		(*n* = 24)	(*n* = 27)	(*n* = 22)	(*n* = 30)	(*n* = 38)	(*n* = 55)	(*n* = 40)	(*n* = 36)
%		8.8	9.9	8.1	11.0	14.0	20.0	14.7	13.2
Gender	Male (*n* = 138)	11	14	3	14	18	24	26	28
	50.7%	(8.0)	(10.1)	(2.2)	(10.1)	(13.0)	(17.4)	(18.8)	(20.3)
	Female (*n* = 134)	13	13	19	16	20	31	14	8
	49.3%	(9.7)	(9.7)	(14.2)	(11.9)	(14.9)	(23.1)	(10.4)	(6.0)
Ward	OPD (*n* = 123)	15	10	7	15	21	24	16	15
	45.2%	(12.5)	(8.1)	(5.7)	(12.2)	(17.1)	(19.5)	(13.0)	(12.2)
	Nephrology (*n* = 58)	0	2	9	9	8	13	8	9
	21.3%	(0.0)	(3.4)	(15.5)	(15.5)	(13.8)	(22.4)	(13.80	(15.5)
	Emergency (*n* = 26)	1	4	1	1	1	9	4	5
	9.6%	(3.8)	(15.4)	(3.8)	(3.8)	(3.8)	(34.6)	(15.4)	(19.2)
	ICU (*n* = 13)	1	4	3	0	0	1	1	3
	4.8%	(7.7)	(30.8)	(23.1)	(0.0)	(0.0)	(7.7)	(7.7)	(23.1)
	Urology (*n* = 52)	7	7	2	5	8	8	11	4
	19.1%	(13.5)	(13.5)	(3.8)	(9.6)	(15.4)	(15.4)	(21.2)	(7.7)
Bacterial isolate	*Escherichia coli* (*n* = 174)	14	14	11	20	28	40	22	25
	64.0	(8.0)	(8.0)	(6.3)	(11.5)	(16.1)	(23.0)	(12.6)	(14.4)
	*Klebsiella spp*. (*n* = 74)	6	8	6	8	6	14	16	10
	27.2	(8.1)	(10.8)	(8.1)	(10.8)	(8.1)	(18.9)	(21.6)	(13.5)
	*Acinetobacter spp*. (*n* = 18)	4	3	4	2	4	1	0	0
	6.6	(22.2)	(16.7)	(22.2)	(11.1)	(22.2)	(5.6)	(0.0)	(0.0)
	Others (*n* = 6)	0	2	1	0	0	0	2	1
	2.2	(0.0)	(33.3)	(16.7)	(0.0)	(0.0)	(0.0)	(33.3)	(16.7)
Sample collection Source	Urine (*n* = 198)	16	17	15	21	31	39	33	26
	72.8%	(8.1)	(8.6)	(7.6)	(10.6)	(15.7)	(19.7)	(16.7)	(13.1)
	Blood (*n* = 17)	1	4	1	1	3	4	1	2
	6.3%	(5.9)	(23.5)	(5.9)	(5.9)	(17.6)	(23.5)	(5.9)	(11.8)
	Pus (*n* = 16)	0	1	2	1	1	6	2	3
	5.9%	(0.0)	(6.3)	(12.5)	(6.3)	(6.3)	(37.5)	(12.5)	(18.8)
	Throat aspiration (*n* = 13)	4	2	1	1	1	1	1	2
	4.8%	(30.8)	(15.4)	(7.5)	(7.7)	(7.7)	(7.7)	(7.7)	(15.4)
	CVC Tip (*n* = 8)	1	0	2	1	2	0	1	1
	2.9%	(12.5)	(0.0)	(25.0)	(12.5)	(25.0)	(0.0)	(12.5)	(12.5)
	Sputum (*n* = 5)	0	1	0	1	0	2	0	1
	1.8%	(0.0)	(20.0)	(0.0)	(20.0)	(0.0)	(40.0)	(0.0)	(20.0)
	PCN (*n* = 4)	0	0	0	2	0	1	1	0
	1.5%	(0.0)	(0.0)	(0.0)	(50.0)	(0.0)	(25.0)	(25.0)	(0.0)
	Bronchial aspiration (*n* = 4)	1	0	0	1	0	0	1	1
	1.5%	(25.0)	(0.0)	(0.0)	(25.0)	(0.0)	(0.0)	(25.0)	(25.0)
	Others (*n* = 7)	1	2	1	1	0	2	0	0
	2.6%	(14.3)	(28.6)	(14.3)	(14.3)	(0.0)	(28.6)	(0.0)	(0.0)

OPD = outpatient, ICU = intensive-care unit, PCN = *Percutaneous nephrostomy*, CVC = central venous catheters.

**Table 2 antibiotics-12-00029-t002:** Antimicrobial susceptibility testing against ESBL-producing *Escherichia coli*, *Klebsiella* species, *Acinetobacter* species and others.

Antibiotics	*Escherichia coli* *n* = 174			*Klebsiella* Species *n* = 74			*Acinetobacter* Species *n* = 18			Others *n* = 6			Total *n* = 272		
	R	I	S	R	I	S	R	I	S	R	I	S	R	I	S
TZP	45	1	128	43	1	30	17	0	1	2	0	4	107	2	163
	(25.9)	(0.6)	(73.6)	(58.1)	(1.4)	(40.5)	(94.4)	(0.0)	(5.6)	(33.3)	(0.0)	(66.7)	(39.3)	(0.7)	3 (59.9)
MEM	33	1	140	46	0	28	17	0	1	2	0	4	98	1	173
	(19.0)	(0.6)	(80.5)	(62.2)	(0.0)	(37.8)	(94.4)	(0.0)	(5.6)	(33.3)	(0.0)	(66.7)	(36.0)	(0.4)	(63.6)
IPM	35	2	137	44	0	30	17	0	1	2	0	4	98	2	172
	(20.1)	(1.1)	(78.7)	(59.5)	(0.0)	(40.5)	(94.4)	(0.0)	(5.6)	(33.3)	(0.0)	(66.7)	(36.0)	(0.7)	63.2)
G	63	1	110	45	0	29	15	0	3	3	0	3	126	1	145
	(36.2)	(0.6)	(63.2)	(60.8)	(0.0)	(39.2)	(83.3)	(0.0)	(16.7)	(50.0)	(0.0)	(50.0)	(46.3)	(0.4)	(53.3)
AK	19	5	150	31	3	40	15	0	3	2	0	4	67	8	197
	(10.9)	(2.9)	(86.2)	(41.9)	(4.1)	(54.1)	(83.3)	(0.0)	(16.7)	(33.3)	(0.0)	(66.7)	(24.6)	(2.9)	(72.4)
NA	156	3	15	64	0	10	17	0	1	5	0	1	242	3	27
	(89.7)	(1.7)	(8.6)	(86.5)	(0.0)	(13.5)	(94.4)	(0.0)	(5.6)	(83.3)	(0.0)	(16.7)	(89.0)	(1.1)	(9.9)
NOR	150	0	24	58	1	15	17	0	1	6	0	0	231	1	40
	(86.2)	(0.0)	(13.8)	(78.4)	(1.4)	(20.3)	(94.4)	(0.0)	(5.6)	(100.0)	(0.0)	(0.0)	(84.9)	(0.4)	(14.7)
CIP	141	0	33	61	0	13	17	0	1	5	0	1	234	0	48
	(81.0)	(0.0)	(19.0)	(82.4)	(0.0)	(17.6)	(94.4)	(0.0)	(5.6)	(83.3)	(0.0)	(16.7)	(82.4)	(0.0)	(17.6)
SXT	146	0	28	64	0	10	17	0	1	4	0	2	231	0	41
	(83.9)	(0.0)	(16.1)	(86.5)	(0.0)	(13.5)	(94.4)	(0.0)	(5.6)	(66.7)	(0.0)	(33.3)	(84.9)	(0.0)	(15.1)
F	30	3	141	45	2	27	15	0	3	3	0	3	93	5	174
	(17.2)	(1.7)	(81.0)	(60.8)	(2.7)	(36.5)	(83.3)	(0.0)	(16.7)	(50.0)	(0.0)	(50.0)	(34.2)	(1.8)	(64.0)
SCF	50	2	122	45	1	28	16	1	1	2	0	4	113	4	155
	(28.7)	(1.1)	(70.1)	(60.8)	(1.4)	(37.8)	(88.9)	(5.6)	(5.6)	(33.3)	(0.0)	(66.7)	(41.5)	(1.5)	(57.0)
FOS	8	0	166	28	4	42	15	1	2	3	0	3	54	5	213
	(4.6)	(0.0)	(95.4)	(37.8)	(5.40	(56.8)	(83.3)	(5.6)	(11.1)	(50.0)	(0.0)	(50.0)	(19.9)	(1.8)	(78.3)

R = resistant, I = intermediate, S = susceptible, NA = nalidixic acid, SXT = co-trimoxazole, CIP = ciprofloxacin, CN = gentamicin, SCF = sulbactam/cefoperazone, TZP = piperacillin/tazobactam, MEM = meropenem, IPM = imipenem, F = nitrofurantoin, AK = amikacin, FOS = fosfomycin.

**Table 3 antibiotics-12-00029-t003:** Distribution of ESBL coding genes among GNB bacteria isolated from clinical specimens.

Bacterial Isolate N = 272		CTX-M *n* = 248 (91.2%)	OXA *n* = 107 (39.3%)	TEM *n* = 168 (61.8%)	SHV *n* = 48 (17.6%)	*p*-Value
*Escherichia coli*	Positive	160 (92.0)	71 (40.8)	105 (60.3)	25 (14.4)	0.00 *
	Negative	14 (8.0)	103 (59.2)	69 (39.7)	149 (85.6)	
*Klebsiella* species	Positive	68 (91.9)	30 (40.5)	46 (62.2)	18 (24.3)	0.00 *
	Negative	6 (8.1)	44 (59.5)	28 (37.8)	56 (75.7)	
*Acinetobacter* species	Positive	16 (88.9)	5 (27.8)	13 (72.2)	4 (22.2)	0.00 *
	Negative	2 (11.1)	13 (72.2)	5 (27.8)	14 (77.8)	
Others	Positive	4 (66.7)	1 (16.7)	4 (66.7)	1 (16.7)	0.00 *
	Negative	2 (33.3)	5 (83.7)	2 (33.3)	5 (83.7)	

* Prevalence of ESBLs coding genes (bla_CTX-M,_ bla_OXA,,_ bla_TEM,_ and bla_SHV_) in *E. coli*, Klebsiella species, Acinetobacter species and others(Salmonella, Stenotrophomonas species and Serratia).

**Table 4 antibiotics-12-00029-t004:** Antibiotic resistance and presence of ESBLs-coding genes among clinical isolates.

Genes	Total N (%)	Non-β-Lactam Antibiotics
*bla_CTX-M_* + *bla_TEM_*	83 (30.5)	-
*bla_CTX-M_*	46 (16.9)	-
*bla_CTX-M_* + *bla_OXA_* + *bla_TEM_*	38 (14.0)	NA, NOR, CIP
*bla_CTX-M_* + *bla_OXA_*	37 (13.6)	-
*bla_CTX-M_* + *bla_TEM_* + *bla_SHV_*	19 (7.0)	SXT
*bla_CTX-M_* + *bla_TEM_* + *bla_OXA_* + *bla_SHV_*	13 (4.8)	NA, NOR, CIP, SXT,
*bla_TEM_*	7 (2.6)	NA, NOR, CIP,
*bla_OXA_*	6 (2.2)	-
*bla_CTX-M_* + *bla_SHV_*	6 (2.2)	NA, NOR, CIP, SXT, F, SCF, FOS
*bla_CTX-M_* + *bla_OXA_* + *bla_SHV_*	6 (2.2)	NA, NOR, CIP, SXT, F
*bla_OXA_* + *bla_TEM_*	5 (1.8)	NA, NOR, CIP, SXT
*bla_OXA_* + *bla_TEM_* + *bla_SHV_*	2 (0.7)	MEM, IPM, G, AK, NA, NOR, CIP, SXT, SCF, F, FOS
Nil	2 (0.7)	NA, NOR, CIP
*bla_SHV_*	1 (0.4)	MEM, IPM, NA, NOR, CIP, SXT, SCF
*bla_TEM_* + *bla_SHV_*	1 (0.4)	MEM, IPM, G, AK, NA, NOR, CIP, SXT, SCF, F, FOS
*bla_OXA_* + *bla_SHV_*	0	-

**Table 5 antibiotics-12-00029-t005:** Primer used for molecular identification of ESBLs by PCR.

Target Gene	Primer Sequence		Product Size	Annealing Temperature (°C)	References
*bla_TEM_*	F	5′-TCAACATTTCCGTGTCG-3′	860	56	[56]
	R	5′-CTGACAGTTACCAATGCTTA-3′			
*bla_CTX-M_*	F	5′-ATGTGCAGYACCAGTAARGT-3′	593	50	[57]
	R	5′-TGGGTRAARTARGTSACCAGA-3′			
*bla_OXA_*	F	5′-ATATCTCTACTGTTGCATCTCC-3′	619	56	[58]
	R	5′-AAACCCTTCAAACCATCC-3′			
*bla_SHV_*	F	5′-AGGATTGACTGCCTTTTTG-3′	392	56	[58]
	R	5′-ATTTGCTGATTTCGCTCG-3′			

## Data Availability

Not applicable.

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
