# Peer review of "Antimicrobial Susceptibility and Genetic Prevalence of Extended-Spectrum β-Lactamases in Gram-Negative Rods Isolated from Clinical Specimens in Pakistan"

_antibiotics, 2022, doi:10.3390/antibiotics12010029_

Round 1
Reviewer 1 Report
Abstract
· Not clear, are the proportion of resistance are for ESBL producing isolates or not? If the isolates are ESBL producers reporting ampicillin, cefotaxime, ceftazidime, amoxicillin clavulanic acid is misleading, please amend this
· The recommendations given is not coming from conclusion please amend based on the results
· Type of the study??
Introduction
· The introduction is too general: Report the epidemiology of ESBL genes focus on the study area. Please show the gaps.
· Review the article: Prevalence of extended-spectrum-β-lactamase-producing Enterobacteriaceae: first systematic meta-analysis report from Pakistan. Antimicrob Resist Infect Control. 2018; 7: 26. 2018 Feb 20. doi: 10.1186/s13756-018-0309-1
· Check reference number 16: Use appropriate reference that shows how AMU select for ESBL genes
· Strong statements have been given in the last paragraph of the introduction with no references to support them. Please provide references
· Refine the aim of the study to show what exactly was done
Methods
· Section 4.1: provide time and duration of the study including the type of the study
· How many samples were processed? How many isolates were obtained? How many gram negatives? The 272 is what proportion of all isolates in the specified time.
· Please give the description of the Hospital? What is the bed capacity? Please refer the SROBE checklist to ensure each section of the manuscript is well written (https://www.strobe-statement.org/checklists/)
· Provide the manufacturer details of each material used (Company, City, Country)
· Not clear regarding sample management: Why urine samples were preserved? For how long were they preserved?
· Why Blood samples were left at room temperature for a week?
· Why only CLED was used for urine samples? Provide references
· Section 4.2: Not clear regarding Blood samples?? These were supposed to be sub-cultured: Please amend this section accordingly
· Section 4.3: the sentence “along with antimicrobial susceptibility testing” is misleading, please amend. Clearly show which discs interpretation was done according CLSI and which as per EUCAST page 10 line 322
· Section 4.4: Rephrase: Start with what discs were tested before interpretation. Use standard names not commercial names of antibiotics discs?? SXT: trimethoprim/sulfamethoxazole. The brackets are confusing directly report antibiotic discs without classes or use table
· Section 4.6: Rephrase the last sentence of the first paragraph. Give references for the methods or provide details: What concentrations of dNTPs? Taq polymerase? Minerals? Etc. Provide the gels image as a figure showing types of ESBL genes based on the fragment size??
Results
· Were ESBL isolates significantly more from males or females? Add this information: Out of males enrolled how many had ESBL? compared to XX from YYY females enrolled???
· The author should provide information of total samples processed from each unit and out of them how many yielded ESBL isolates. Revise Table 1 or add another table or modify table 2, add information as advised then calculate p value for each variable (Sex, Ward, Isolates etc). For the isolates, include how many E. coli were isolated, then 174/xxx, % ESBL
· Revise section 2.2: Report susceptibility patterns. The word effectiveness is confusing. Here report the overall susceptibility patterns for all isolates, then add a row for MDR, ESBL. Change section 2.3, 2.4 and 2. 5 accordingly
· Table 2: Report the susceptibility for species with more than 30 isolates. Report all isolates including non-ESBL if you want to report only ESBL exclude antibiotics such as: AMP, AMC, CAZ and CTX
· Table 3: What does p value refers to? Is it differences on OXA, CTX-M, TEM?? Please provide information regarding the comparison of ESBL genes by species
· Table 4: revise the table to show what addition resistance were present in in each ESBL gene or combination focus on CIP, CN, SXT, NOR, F
Discussion
· Revise the discussion accordingly. What did you expect?
· Avoid much repetitions of the results, please follow the STROBE guidelines
· Revise the conclusion is ESBL prevalence high?? No data to support this unless the suggested information is added. What is obvious is high CTX-M-alleles??
Author Response
|
S.No |
Question/Query |
Answer/comment |
|
1 |
Abstract 1. Not clear, are the proportion of resistance are for ESBL producing isolates or not? If the isolates are ESBL producers reporting ampicillin, cefotaxime, ceftazidime, amoxicillin clavulanic acid is misleading, please amend this
|
1. We identified 272 isolates of ESBLs phenotypically among which 02 isolates were negative to ESBLs coding genes. So, we amended this accordingly as: “These ESBLs-producing isolates shown resistance to β-lactam antibiotics i.e. ampicillin (100%), amoxicillin/clavulanate (99.3%), cefotaxime (99.3%), ceftazidime (99.3%) and non- β-lactam antibiotics i.e. nalidixic acid (89.0%), co-trimoxazole (84.9%), ciprofloxacin (82.4%), gentamicin (46.3%), sulzone (41.5%), piperacillin/tazobactam (39.3%), meropenem (36.0%), imipenem (34.2%), nitrofurantoin (24.6%), amikacin (19.9%) and fosfomycin (19.9%), respectively”. |
|
2. The recommendations given is not coming from conclusion please amend based on the results.
|
2. The recommendation has been adopted in the revised manuscript based on our results as; “ESBLs producing GNRs showed resistance to β-lactam antibiotics associated with blaCTX-M, blaTEM, blaOXA, and blaSHV were still shown susceptibility to β-lactam inhibitor and non-β-lactam antibiotics i.e. aminoglycoside (amikacin and gentamicin), carbapenem (meropenem and imipenem), nitrofurantoin and fosfomycin that can be used as a choice of drug against infectious disease in our clinical settings”.
|
|
|
3. Type of the study??
|
3. We described the type of study as: “This cross-sectional study aimed to determine the antimicrobial susceptibility in ESBLs producing GNRs and its correlation with corresponding genes.” |
|
|
2 |
Introduction 1. The introduction is too general: Report the epidemiology of ESBL genes focus on the study area. Please show the gaps.
|
1. Thanks for your valuable remarks. We addressed this suggestion in Introduction as: “World Health Organization (WHO) has showed his serious concern about hike of antibiotic resistance in developing countries especially Pakistan and asked to overcome this issue [2]. Different authors conducted their studies in different regions of Pakistan [23] as Lahore (2022) [3], Timurgara [4] and Islamabad [5] but, no study was conducted in Multan located in South Punjab, Pakistan. That’s why, the importance of this study has also been increased due to lack of epidemiological data and prevalence of ESBLs producer and its antimicrobial susceptibility patten in our population”. |
|
2. Review the article: Prevalence of extended-spectrum-β-lactamaseproducing Enterobacteriaceae: first systematic meta-analysis report from Pakistan. Antimicrob Resist Infect Control. 2018; 7: 26. 2018 Feb 20. doi: 10.1186/s13756-018-0309-1
|
2. Thank you very much for providing us support to make introduction fruitful. The mentioned reference at place 23 is added and discussed in revised manuscript.
|
|
|
3. Check reference number 16: Use appropriate reference that shows how AMU select for ESBL genes
|
3. We updated the reference in revised introduction as: “Overuse of antibiotics, inappropriate hygienic practices in clinical setting and lack of continuous monitoring of drug resistance pattern have provided an appropriate environment for emergence and uncontrolled dissemination of AmpC and ESBLs enzymes 1. |
|
|
4. Strong statements have been given in the last paragraph of the introduction with no references to support them. Please provide references
|
4. We provided the reference in last paragraph of Introduction as: “It is an alarming situation for clinical setting to treat and overcome the infectious disease caused by ESBLs producing pathogens leading to resistance against commonly used antibiotics. There is a lack of awareness in both population and health care workers about their prevalence and potential side effects to control the infectious diseases (20) The incidence of ESBLs producing pathogens is tremendously increasing due to lack of proper laboratory diagnosis, proper reporting and preventive measures in our clinical settings (21,22).” |
|
|
5. Refine the aim of the study to show what exactly was done |
6. We updated the aim of the study in revised transcript as: “In view of this, the study was held to find out the most susceptible antibiotics that can be used as a choice of drug against ESBLs producing isolates and the prevalence of corresponding genes (blaCTX-M, blaOXA, blaTEM and blaSHV) of ESBLs among clinical isolates in Multan Institute of Kidney Disease Hospital, Multan, Pakistan”.
|
|
|
3 |
Methods 1. Section 4.1: provide time and duration of the study including the type of the study ·
|
. 1. We mentioned the time and type of study in section 4.1 as: “This cross-sectional study was designed and conducted at Bahauddin Zakariya University, Multan from September 2020 to March 2021 and ethical approval was granted by the Institutional Review Board (IRB) of IMBB (Reference No. 334/A)”.
|
|
2. How many samples were processed? How many isolates were obtained? How many gram negatives? The 272 is what proportion of all isolates in the specified time. |
2. This study was aimed & was focused only on ESBLs producing bacteria. We did not collected data of other isolates or no of samples analyzed. |
|
|
3. Please give the description of the Hospital? What is the bed capacity? Please refer the SROBE checklist to ensure each section of the manuscript is well written (https://www.strobestatement.org/checklists/)
|
3. Now, we mentioned the hospital details in section 4.1 according to the reviewers suggestion. “Samples were collected from patients admitted in different wards i.e., out-patient door (OPD), Emergency, Nephrology, Urology and intensive care unit (ICU) of Multan Institute of Kidney Diseases Hospital, Multan, Pakistan (150-bed and single specialty hospital) without age and gender discrimination.” |
|
|
4. Provide the manufacturer details of each material used (Company, City, Country) |
4. We mentioned details according to the reviewer suggestions: “Urine, sputum, fluids and PCN samples were collected in sterile container and blood samples in culture vial (BD BACTEC™ Plus Aerobic/F, New Jersey, United States)” |
|
|
5. Not clear regarding sample management: Why urine samples were preserved? For how long were they preserved?
|
5. The storage of processed samples must include sufficient time to resolve any type of complaint in analysis of samples. Then, we may be able to address any request by the other clinician or the microbiologist who is involved in processing the samples. That’s why sample preserved for one week. |
|
|
6. Why Blood samples were left at room temperature for a week?
|
6. We followed the instruction as given reference as “Samples were transported to the laboratory after collection by following different instructions as urine sample within 2 to 4 hours on ice pack and blood culture vial (BD BACTEC™ Plus Aerobic/F, New Jersey, United States) 4 to 8 hours at room temperature. Urine samples were preserved at 2°C to 8°C in refrigerator and blood culture vials at room temperature for one week 5. |
|
|
7. Why only CLED was used for urine samples? Provide references
|
7. According to “District laboratory practice in tropical countries, part 2, page number 114”, only CLED is used for urine samples. Reference no. 50 and line 278 is provided. |
|
|
8. Section 4.2: Not clear regarding Blood samples?? These were supposed to be sub-cultured: Please amend this section accordingly
|
8. We amended this as: “After inoculation, plates were placed at 37°C in incubator for overnight incubation. In next day (after 24 hours), plates were checked for bacterial growth and sub-cultured pure growth for making bacterial glycerol stock, gram staining, biochemical analysis for bacterial identification, antimicrobial susceptibility testing and DNA extraction”.
|
|
|
9. Section 4.3: the sentence “along with antimicrobial susceptibility testing” is misleading, please amend. Clearly show which discs interpretation was done according CLSI and which as per EUCAST page 10 line 322
|
9. Thanks for pointing out this, actually, both have same guidelines except few that were not implemented in this study see the following explanation in text: “After confirmation of gram-negative rods, analytical profile index 20 Enterobacteriaceae (API 20E, bioMérieux, Durham, USA) was used to characterize the bacterial strain as Escherichia coli, Klebsiella species, Acinetobacter species, Proteus species, Serratia species and Salmonella species 6 followed by antimicrobial susceptibility testing that was used to find the resistant and susceptible antibiotics.” |
|
|
10. Section 4.4: Rephrase: Start with what discs were tested before interpretation. Use standard names not commercial names of antibiotics discs?? SXT: trimethoprim/sulfamethoxazole. The brackets are confusing directly report antibiotic discs without classes or use table
|
10. Thanks for this nice suggestion and the name of antibiotics were changed according to the reviewer remarks.
|
|
|
11. Section 4.6: Rephrase the last sentence of the first paragraph. Give references for the methods or provide details: What concentrations of dNTPs? Taq polymerase? Minerals? Etc.
|
11. We rephrased the last sentence of first paragraph as: “1% agarose gel containing ethidium bromide was used to confirm the DNA extraction under UV light”. Secondly, we used ready-to-use master mix (Vazyme Biotech co., Nanjing, China and Catalog # P311, P312). We did not prepare master mix manually and reference is given.
|
|
|
12. Provide the gels image as a figure showing types of ESBL genes based on the fragment size??
|
12. We inserted the gel images in section 4.6. Figure 1. Visualization of DNA bands by gel electrophoresis. (a) Lane-NC and PC are for Negative and Positive control, respectively. Lane-12 is showing band of 392bp which is related to SHV gene and lane-1, 2, 3, 4, 5, 6, 7, 8, 9, 10 and 11are negative for SHV gene (b) Lane-NC and PC are for Negative and Positive control, respectively. Lane-1, 2, 3, 4 and 6 are showing band of 860 bp which is related to TEM gene. Lane-7 and 10 are showing band of 619 bp which is related to OXA gene. Lane-8, 9 and 11 are showing both 860 bp and 619 bp bands which are related to TEM and OXA, respectively. Lane-5 has no band (c) Lane-NC and PC are for Negative and Positive control, respectively. Lane-1 to 12 are showing band of 593 bp which is related to CTX-M gene. |
|
|
4 |
Results 1. Were ESBL isolates significantly more from males or females? Add this information: Out of males enrolled how many had ESBL? compared to XX from YYY females enrolled??? |
1. We mentioned this in Table 1 and text Line 94-95 and prevalence of ESBLs in males is 50.7% and 49.3% in female.
|
|
2. The author should provide information of total samples processed from each unit and out of them how many yielded ESBL isolates. Revise Table 1 or add another table or modify table 2, add information as advised then calculate p value for each variable (Sex, Ward, Isolates etc). For the isolates, include how many E. coli were isolated, then 174/xxx, % ESBL |
2. Thank you very much for highlighting this issue. We focused on ESBLs producing bacteria in this study. We did not collect data of all samples except ESBLs producing bacteria. we don’t have this data at this movment.
|
|
|
3. Revise section 2.2: Report susceptibility patterns. The word effectiveness is confusing. Here report the overall susceptibility patterns for all isolates, then add a row for MDR, ESBL. Change section 2.3, 2.4 and 2. 5 accordingly |
3. We replaced effectiveness with susceptibility. we had no data of overall susceptibility against pathogens.
|
|
|
4. Table 2: Report the susceptibility for species with more than 30 isolates. Report all isolates including non-ESBL if you want to report only ESBL exclude antibiotics such as: AMP, AMC, CAZ and CTX |
4. According to CLSI guideline, we used these antibiotics against Enterobacteriaceae and also helped us to identify the ESBLs phenotypically.
|
|
|
5. Table 3: What does p value refers to? Is it differences on OXA, CTX-M, TEM?? Please provide information regarding the comparison of ESBL genes by species
|
5. We mentioned it in the legend of Table 3 as “Prevelance of ESBLs coding genes (blaCTX-M, blaOXA,, blaTEM, and blaSHV) were associated significantly with E. coli, Klebsiella species, Acinetobacter species and others (Salmonella, Stenotrophomonas species and Serratia)”. |
|
|
6. Table 4: revise the table to show what addition resistance were present in in each ESBL gene or combination focus on CIP, CN, SXT, NOR, F |
6. We revised and tried to simplify the Table 4 for the understanding of reader. |
|
|
|
Discussion 1. Revise the discussion accordingly. What did you expect? Avoid much repetitions of the results, please follow the STROBE guidelines |
1. We try revised as mentioned by reviewer, hopefully, it will be up to the mark now. |
|
2. Revise the conclusion is ESBL prevalence high?? No data to support this unless the suggested information is added. What is obvious is high CTX-M-alleles?? |
2. We revised the conclusion as “In conclusion, blaCTX-M were the most prevalent gene coding to ESBLs production among E. coli, Klebsiella species, Acinetobacter species and other clinical isolates followed by blaTEM, blaOXA and blaSHV”. |

Reviewer 2 Report
The authors analyzed the clinical data they had collected regarding the bacteria, ESBL antimicrobial profile and genes from 272 samples. This is a study that reports the statistics of clinical results. It is useful to show where is the main problem with ESBL producing bacteria. The manuscript is easy to read, although several minor typos are found throughout the text. The authors need to revise carefully.
I would use the term Gram-negative rod-shaped bacteria instead of Bacilli, because the later term refers to a Gram-positive genus.
A supplementary excel file is needed, where each sample and its accompanying information including bacteria, antibiotic resistance profiles will be recorded, in order for other researchers to make good use of this study.
I think it would benefit the manuscript to briefly mention in the Discussion the new trends in clinical Microbiology, related to antimicrobial resistance profiles, that include genome sequencing followed by in-silico AMR profile prediction (DOI: 10.3390/microorganisms10051040, doi: 10.1016/S2666-5247(21)00117-8, DOI: 10.1038/s41467-021-23091-2). It would also be useful to discuss that genomic analyses show that the most frequent number of antimicrobial resistance genes in a bacterial pathogen is two (DOI: 10.3390/microorganisms10051040).
Line 39: rod-shaped bacteria… that cause resistance against
Line 169: resistance
Line 173: United States
Line 198: The this study
Author Response
|
S.No. |
Question/Query |
Answer/comment |
|
1 |
Comments and Suggestions for Authors 1. The authors analyzed the clinical data they had collected regarding the bacteria, ESBL antimicrobial profile and genes from 272 samples. This is a study that reports the statistics of clinical results. It is useful to show where is the main problem with ESBL producing bacteria. The manuscript is easy to read, although several minor typos are found throughout the text. The authors need to revise carefully. |
1. We revised the paper according to the suggestion of reviewer. Hopefully, it will be improved now in revised version.
|
|
2. I would use the term Gram-negative rod-shaped bacteria instead of Bacilli, because the later term refers to a Gram-positive genus. |
2. According to the reviewer’s suggestion, we replaced bacilli with rods. |
|
|
3. A supplementary excel file is needed, where each sample and its accompanying information including bacteria, antibiotic resistance profiles will be recorded, in order for other researchers to make good use of this study. |
3. We inserted mandatory data in file and uploaded with this revision.
|
|
|
4. I think it would benefit the manuscript to briefly mention in the Discussion the new trends in clinical Microbiology, related to antimicrobial resistance profiles, that include genome sequencing followed by in-silico AMR profile prediction (DOI: 10.3390/microorganisms10051040, doi: 10.1016/S2666- 5247(21)00117-8, DOI: 10.1038/s41467-021-23091-2). It would Y  also be useful to discuss that genomic analyses show that the most frequent number of antimicrobial resistance genes in a bacterial pathogen is two (DOI: 10.3390/microorganisms10051040). |
4. Thank you very much for your assistance to make discussion fruitful and mentioned these reference in revised manuscript as: "Gene sequencing may also provide the phylogenetic history that will play a vital role to overcome the further dissemination of ESBLs corresponding genes (23,47): line 240-242.
|
|
|
5. Line 39: rod-shaped bacteria… that cause resistance against
|
5. Thank you and made amendments in line 39 as: “The β-lactamases are mainly plasmid-mediated enzymes produced by some rod shaped bacteria that cause antimicrobial resistance against……..” |
|
|
6. Line 169: resistance Line
|
6. We revised according to reviewer’s suggestion in line 169 as “Our study is reporting a high level of antibiotic resistance against β-lactam antibiotics including ……” and line 187 now. |
|
|
7. 173: United States |
7. We corrected this as “United States in line 191 now” |
|
|
8. Line 198: The this study |
8. Thanks for this correction, we modified line 216 as “This study, the molecular prevalence” |

Reviewer 3 Report
In this paper, the authors focus on the extended-spectrum β-lactamases (ESBLs) and describe the antimicrobial susceptibility and some genetic existence among the clinical isolates. The authors do not demonstrate the in-depth hypothesis or mechanism about the effectiveness of different antibiotics against different gram-negative species, especially for the difference between β-lactam antibiotics and other kinds of antibiotics. The correlation between antibiotic resistance and ESBL genes lacks more specific or detailed evidence.
Line 51-55, relevant references are needed here for the introduction of different types of β-lactamases.
Line 85, the full name of “OPD” is required for the 1st time shown in the manuscript, although it’s shown in the Materials and Methods later.
Line 96-99, it’s confusing about how the resistant and effective drugs rank here. The authors need to give an explanation or definition first, based on percentage of “R”, “I” or “S”? The same as the rank below.
Line 132-134, it’s better to put the full names of “R”, “I”, “S” in the legend of Table 2.
Line 142, the meaning of p-value is not clear in Table 3.
In the method and result part, the authors used disk diffusion to do the antimicrobial activity. Is it about the measurement of zone inhibition or something else? This needs to be clarified.
Author Response
|
S.No. |
Question/Query |
Answer/comment |
|
1 |
Comments and Suggestions for Authors 1. In this paper, the authors focus on the extended-spectrum β-lactamases (ESBLs) and describe the antimicrobial susceptibility and some genetic existence among the clinical isolates. The authors do not demonstrate the in-depth hypothesis or mechanism about the effectiveness of different antibiotics against different gram-negative species, especially for the difference between βlactam antibiotics and other kinds of antibiotics. The correlation between antibiotic resistance and ESBL genes lacks more specific or detailed evidence. |
1. We appreciate a highly relevant and valid comments related to this study. This study aim was only focused on the ESBL producing bacteria and we did not collected data on other bacterial isolates. So, a detailed correlations analysis of different antibiotic resistance with ESBL genes were lacking, bacause the antibiotic resistance was very obvioius with no. of resistant bacteria (almost 100%) in these isolates, those were having ESBL associated genes. Therefore, we did not calculate or mentione correlations. However, this study is suggesting a clear and detail evience or link between antibiotic resistance and the presence of ESBL genes. |
|
2. Line 51-55, relevant references are needed here for the introduction of different types of β-lactamases. |
2. Relevent references are addad according to the suggestion of the reviewers. (see ref. 2,3,4,5 & 23 in text) |
|
|
3. Line 85, the full name of “OPD” is required for the 1 time shown in the manuscript, although it’s shown in the Materials and Methods later. |
3. We entered the full name of OPD as “Prevalence of ESBLs were more abundant in out-patient-door (OPD) patients 45.2% (n=123) followed by…..” in text and line 99. |
|
|
4. Line 96-99, it’s confusing about how the resistant and effective drugs rank here. The authors need to give an explanation or definition first, based on percentage of “R”, “I” or “S”? The same as the rank below. |
4. Thank you very much for highlighting this issue. We ranked this data according to Table 2.
|
|
|
5. Line 132-134, it’s better to put the full names of “R”, “I”, “S” in the legend of Table 2. |
5. Thank you very much. Now, we mentioned the full name of R, I and S in the legend. |
|
|
6. Line 142, the meaning of p-value is not clear in Table 3. |
6. Thank you very much. We explained it in legend of Table 3 as “Prevelance of ESBLs coding genes (blaCTX-M, blaOXA,, blaTEM, and blaSHV) were associated significantly with E. coli, Klebsiella species, Acinetobacter species and others(Salmonella, Stenotrophomonas species and Serratia)”. |
|
|
7. In the method and result part, the authors used disk diffusion to do the antimicrobial activity. Is it about the measurement of zone inhibition or something else? This needs to be clarified. |
7. We mentioned it clearly in the text now. “Kirby-bauer Disk diffusion method was used to determine the antimicrobial activity by measuring zone of inhibition. In this study, 0.5 McFarland was used to make inoculum by mixing bacterial colonies and normal saline in a test tube. This suspension was spread on Muller Hinton agar (MHA, Oxoid, Basingstoke Hampshire, United Kingdom) plates by cotton swab and then, following antibiotic disks (Oxoid, Basingstoke Hampshire, United Kingdom) were dispensed on it. Zone of inhibition was measured and interpreted into Resistant (R), Intermediate (I) and Sensitive (S) according to…………” |

Round 2
Reviewer 1 Report
· There is no need to report these antibiotics for ESBL producing isolates ampicillin (100%), amoxicillin/clavulanate (99.3%), cefotaxime (99.3%), ceftazidime (99.3%)
· Only piperacillin/tazobactam was tested with resistance of 39.3%, therefore the conclusion regarding Inhibitor is wrong please revise this
· Regarding the total samples analyzed to obtain 273 isolates can be obtained from the lab, this will add value in the interpretation of the data
· Revise the section to show the samples were preserved after processing
· Revise regarding Blood culture: The blood culture bottles were incubated for 24hrs followed by subculture………………
Author Response
|
S.No |
Question/Query |
Answer/comment |
|
1 |
1. There is no need to report these antibiotics for ESBL producing isolates ampicillin (100%), amoxicillin/clavulanate (99.3%), cefotaxime (99.3%), ceftazidime (99.3%). |
1. We revised the paper according to the suggestion of the reviewer. We deleted mentioned antibiotics in the abstract, result section, table 2.
|
|
2. Only piperacillin/tazobactam was tested with resistance of 39.3%, therefore the conclusion regarding Inhibitor is wrong please revise this
|
2. We mentioned the name of the inhibitor antibiotic in the revised conclusion accordingly as: “ESBLs producing GNRs showed resistance to β-lactam antibiotics associated with blaCTX-M, blaTEM, blaOXA, and blaSHV were still shown susceptibility to β-lactam inhibitor (piperacillin/tazobactam) and non-β-lactam antibiotics i.e. aminoglycoside (amikacin and gentamicin), carbapenem (meropenem and imipenem), nitrofurantoin and fosfomycin that can be used as a choice of drug against infectious disease in our clinical settings” |
|
|
3. Regarding the total samples analyzed to obtain 273 isolates can be obtained from the lab, this will add value in the interpretation of the data |
3. We agreed with the suggestion of the reviewer. We appreciated very positive contribution, but unfortunately, in this approved study, we did not have access to all types of samples and lab. data. The aim of this approved study, access to the lab. and samples with patient data were limited. We were able to manage only these 273 isolated, with the specific aim of this study. We do not have any other data except the excel sheet of submitted previous in supplementary data. |
|
|
4. Revise the section to show the samples were preserved after processing |
4. We revised this section according to the reviewers comments as “After processing, Urine samples were preserved at 2°C to 8°C in refrigerator and blood culture vials at room temperature for one week” |
|
|
5. Revise regarding Blood culture: The blood culture bottles were incubated for 24hrs followed by subculture |
5. We revised this according to reviewer’s suggestions as: “The blood culture bottles were incubated at room temperature for 24 hours followed by subculture onto blood, chocolate and MacConkey agar.” |

Reviewer 2 Report
The authors have addressed my comments/corrections.
Author Response
We are thankful to the reviewer for the acceptance of our revisions and point-by-point explanations.

Reviewer 3 Report
The revised version looks good.
Author Response

(The authors gave the same response as above.)
